# Microservice Application Scheduling in Multi-Tiered Fog-Computing-Enabled IoT

**DOI:** 10.3390/s23167142

**Published:** 2023-08-12

**Authors:** Maria Ashraf, Muhammad Shiraz, Almas Abbasi, Omar Alqahtani, Gran Badshah, Ayodele Lasisi

**Affiliations:** 1Department of Computer Science, Faculty of Computing and Information Technology, International Islamic University, Islamabad 44000, Pakistan; almas.abbasi@iiu.edu.pk; 2Department of Computer Science, Federal Urdu University of Arts, Science and Technology, Islamabad 44000, Pakistan; muhammad.shiraz@fuuast.edu.pk; 3Department of Computer Science, King Khalid University, Abha 61413, Saudi Arabia; osalqahtani@kku.edu.sa (O.A.); gdostan@kku.edu.sa (G.B.); alasisi@kku.edu.sa (A.L.)

**Keywords:** fog computing, constrained devices, Internet of Things, microservice application scheduling, service delay, distributed application execution

## Abstract

Fog computing extends mobile cloud computing facilities at the network edge, yielding low-latency application execution. To supplement cloud services, computationally intensive applications can be distributed on resource-constrained mobile devices by leveraging underutilized nearby resources to meet the latency and bandwidth requirements of application execution. Building upon this premise, it is necessary to investigate idle or underutilized resources that are present at the edge of the network. The utilization of a microservice architecture in IoT application development, with its increased granularity in service breakdown, provides opportunities for improved scalability, maintainability, and extensibility. In this research, the proposed schedule tackles the latency requirements of applications by identifying suitable upward migration of microservices within a multi-tiered fog computing infrastructure. This approach enables optimal utilization of network edge resources. Experimental validation is performed using the iFogSim2 simulator and the results are compared with existing baselines. The results demonstrate that compared to the edgewards approach, our proposed technique significantly improves the latency requirements of application execution, network usage, and energy consumption by 66.92%, 69.83%, and 4.16%, respectively.

## 1. Introduction

With the proliferation of smart mobile devices (SMDs), ubiquitous computing, and advances in the IoT, diverse real-time applications can be executed on a plethora of these devices using their available idle resources. Emerging real-time IoT applications and cyber physical systems (CPSs) include real-time traffic monitoring, object tracking using portable video devices, health monitoring and autonomous vehicles, to name a few. Integration and collaboration of these applications from diverse domains lead to the vision of smart sustainable cities that can solve complex scenarios and situations for the development of sustainable smart IoT systems [1].

The cloud outsources significant computational and storage services to end users and makes it possible to execute computationally intensive applications on these resource-hungry devices [2]. However, high latencies that result from using central cloud services (CDCs) cannot be tolerated for real-time applications [3]. The emergence of edge/fog computing facilitates the provision of cloud services at the network edge, close to end users. This arrangement reduces latencies and bandwidth requirements on the communication infrastructure [4]. Fog computing (FC) is a hierarchical architecture that provides a low latency for the running of resource-hungry applications.

Microservice design allows for the development and deployment of applications as a collection of small, loosely coupled modular components, commonly referred to as ‘microservices’. These microservices communicate through lightweight protocols to efficiently execute services requested by end users [5,6]. These services are features of an IoT application that a user may request. Nowadays, a growing number of influential IT firms or application providers, including Amazon, Netflix, and the Guardian, are developing large and complex applications with the help of microservice technologies (e.g., Kubernetes, Apache Mesos, etc.) [7]. The benefits of breaking up applications into a number of functionally distinct but logically related microservices include an increase in flexibility and ease of updating, but it also necessitates more effort from the developers to handle external complexities in application development, communication control, and failure recovery.

This has resulted in the deployment of applications on the fog or edge computing paradigm in a distributed manner [8]. The fog nodes (FNs) have heterogenous computing and storage resources and end users have different application preferences. To achieve efficient application execution on resource-constrained devices, the following key issues must be addressed:Given diverse FNs are available along the path from end users to the central cloud, how can we effectively select the most suitable FN or edge node (EN) by leveraging its idle resources and efficiently schedule microservices on the appropriate nodes?Sending services to a server for remote execution and receiving the results can further increase the overall delay of application execution. How can the latency of such critical applications be minimized?

To tackle the aforementioned challenges, we have designed an enhanced technique for microservice application placement in a multi-tiered fog-to-cloud architecture. For brevity, we call this dynamic microservice application placement (DMAP). This technique focuses on optimizing the overall latency of application execution. It leverages idle resources present in nearby IoT devices such as laptops and mobile phones, in addition to FNs and cloud servers. In this research, the primary focus is on the deployment challenge of microservice-based applications, considering the average latency and resource utilization of the edge network. The main contributions of this work are summarized as follows:We propose utilizing resources at hand from nearby IoT devices, FNs, and CDCs to establish an efficient, multi-tiered fog-to-cloud architecture. The primary objective of this architecture is to allocate services in proximity to end users whenever possible, with the goal of reducing latency, energy consumption, and network usage.We propose a decentralized resource mapping algorithm to allocate resources in a multi-tiered fog-to-cloud architecture. The algorithm aims to identify the optimal mapping of resources to tasks, ensuring an appropriate allocation.Finally, we demonstrate the performance of the proposed technique through simulations. Moreover, we also investigate the tradeoffs of shifting microservices from the traditional computing model of the fog–cloud and the results are compared with existing methods based on latency, bandwidth usage, and energy consumption.

The rest of this paper is organized as follows. Section 2 presents the related work on application placement. Section 3 introduces a description of the multi-tier fog-to-cloud architecture while illustrating the offloading problem. Section 4 presents the proposed technique. In Section 5, the proposed technique is investigated and elaborated on. Finally, the conclusions and future directions are highlighted in Section 6.

## 2. Related Work

In this section, we highlight the benefits of challenges introduced by distributed application execution in an FC environment, followed by a discussion of the need for application scheduling in the multi-tiered fog-to-cloud architecture that is being proposed in this paper.

### 2.1. Application Scheduling in Multi-Tiered Fog-to-Cloud Architecture

Numerous strategies are proposed in existing research to schedule applications in fog environments [9,10,11,12]. Some studies consider full application placement strategies [13], while others consider application service placement in which modules are placed on the fog servers so that multiple end users can use them [14,15].

Initially, Deng et al. [7] proposed a three-layer fog-to-cloud architecture to efficiently utilize resources. They proposed two algorithms for resource scheduling; first fit and random fit in a two-layer fog-to-cloud environment. Ren et al. [13] proposed improved resource scheduling to handle delay sensitive applications in three-layer fog-to-cloud architecture according to the delay sensitivity of the applications.

Skarlat et al. [16] introduced the concept of fog colonies. They considered one fog colony, and services were offloaded to the cloud if there was no host available to deploy the services in the colony. They extended their work and proposed if the host is not available for service deployment, resources in neighboring fog colonies should be considered [17]. Table 1 presents a comparison between the existing application placement strategies and our proposed approach DMAP.

### 2.2. Discussion/Comparison

Efficiently utilizing fog and cloud resources and enhancing the quality of service during application execution requires that the crucial matter of resource scheduling be addressed. Most of the schemes consider two-tier fog-to-cloud architecture; however, they are not effective in a multi-tiered FC environment. The challenges caused by significant delays and network utilization in cloud services, as well as the opportunity to better utilize idle resources in low-resource (L-R) ENs/FNs belonging to nearby end users, prompted us to examine resource mapping. This motivated us to look into how we may utilize the resources available in the L-R edge tier. This paved our way to proposing a decentralized scheduling in FC for delay-sensitive applications, which we will discuss in detail in the following section.

To summarize, we can conclude that existing literature has shown the potential of IoT and FC concepts in facilitating real-time application execution and addressing the difficulties associated with remote CDCs. Nevertheless, a significant portion of these studies did not provide a demonstration of the selection and effective management of available and suitable resources through L-R FNs to further minimize response time and reduce the energy consumption associated with data transfer to remote FNs. The main motivation of this work is to develop a distributed microservices scheduling solution for multi-tiered fog computing-enabled IoT, as presented in Section 3.

## 3. System Model

We consider an environment consisting of multiple IoT devices, an n-tier edge to FC with heterogenous servers, and multiple cloud servers, as shown in Figure 1. We consider microservices-based application workflows that can be represented as a Directed Acyclic Graph (DAG) [11].

### 3.1. Microservice-Based Applications

Microservices architecture is used in the design and development of emerging IoT applications to facilitate robust changes and the agile development. The microservices architecture provides increased granularity, allowing for the composition of multiple microservices within a single service. These microservices collaborate with each other to fulfill end-user requests. Various services may use a unique microservice. Therefore, it is more flexible and agile to define QoS settings at the composite service level as opposed to the application level. A microservices-based application is represented using a DAG, in which the vertices represent individual microservices and the edges represent the data dependencies among these microservices.

### 3.2. Microservices Application Scheduling

Our proposed approach DMAP aims to satisfy the critical latency requirements of application execution by using the underutilized resources of nearby devices instead of placing services on the cloud data centers in a multi-tiered FC environmment. Idle resources of smart IoT devices can also be utilized to provide the computational services. In DMAP, traditional fog devices comprise resource-rich (R-R) fog-tier and SMDs or IoT devices with processing capability and storage availability form the L-R edge-tier. These tiers can be extended to an n-tier fog cloud computing environment to reduce latency, energy consumption and bandwidth usage in an IoT environment, hence improving the QoS of application execution [7]. The higher the FN is from the IoT device, the more computational resources it contains. A three-tier fog-to-cloud architecture, along with its connection, is shown in Figure 2. It can easily be extended to an n-tier FC architecture.

The main components in three-tier fog-to-cloud computing architecture are:IoT Device/Things: IoT devices or other SMDs, e.g., heart rate monitors, can communicate to transmit the sensed data to fog tier using the following protocols: IEEE 802.15.4, WiFi, Bluetooth, MQTT, etc., for data processing and information retrieval. They include mobile phones, laptops, computers, cameras, smartwatches, etc., as well as small energy and power-constrained sensors and actuators.L-R Edge: Represents L-R edge devices that are present next to IoT devices. They are responsible for processing data sensed by the things tier. They are equipped with computational, storage and communicational capabilities and are present between R-R fog servers and IoT devices. Every fog device consists of a tier controller that receives tasks from low-tier devices. These tasks, along with their CPU and delay requirements, are sent to an appropriate device for execution.R-R Fog: consists of fog servers that are more powerful resources as compared to the local edge, and they act as a bridge to connect to CDC.Cloud: provides ubiquitous access to huge shared resources (e.g., computational and storage) over the network.Tier Controller: It makes the decision whether to admit the task or offload it to a higher tier. Tasks are inserted into the queue on arrival. Highly sensitive tasks are sent to the L-R edge tier for execution. If the resources are available, they are executed there; otherwise, they are sent to the controller at R-R fog-tier. Delay-tolerant tasks, however, are sent to a conventional cloud tier for remote execution.

In order to execute applications on resource constraint devices, the delay-sensitive tasks are offloaded to nearby L-R or R-R fog tier depending on the resource availability of ubiquitous mobile devices in the FC environment. We propose a priority rule: if sufficient computing resources are available at the L-R edge-tier, it is offloaded. Otherwise, task placement is shifted upwards towards the R-R fog-tier on the way towards the cloud. If the available device has sufficient resources and can share its computing services, the task is offloaded there; otherwise, the task is executed on a central cloud.

When a mobile device joins a network, it connects with the access point (AP) in the L-R edge-tier. Whenever an application requests computational services for execution of a service, the request is sent to the tier controller, along with the required amount of CPU resource, memory, and its delay requirements. The tier controller resides within the broker and serves the purpose of granting access to services on devices located in L-R, R-R, or the conventional cloud. This access is determined based on the resource requirements of the task and the resources that are currently available on the fog device within L-R, R-R, or the conventional cloud, ensuring their suitability for execution. If the processing requirement of the task is greater than the processing capabilities of the fog device, the task is shifted upwards towards the cloud, where the resources are already sufficient to fulfill the processing requirements of the task.

The tier controller, as shown in Figure 3, is responsible for resource allocation. The data produced can be processed at EN, FN or the cloud depending on the service requirement and the resource capabilities of the computing node. The task scheduler is responsible for actual scheduling of tasks on the virtual machine at the fog server. If a high number of requests is received by tier controller, the requests are shifted upwards towards the cloud to release the pressure on L-R ENs and to ensure efficient resource utilization in fog-to-cloud architecture. The scenario in which an intelligent decision is also made to choose an appropriate server in a fog-to-cloud architecture for task execution will be considered in our future work.

### 3.3. Performance Metric

We formulate the task offloading problem as an optimization problem that aims to minimize latency, network usage and service failure rate. The proposed algorithm is presented in Section 4. To evaluate the performance of the proposed approach, the following performance metrics are considered.

#### 3.3.1. Minimize Delay

The total delay of microservice, denoted by, *m* [25] can be measured as follows: (1)$$D_{m}\left(t\right)=C_{m}\left(t\right)+T_{m}\left(t\right)+W_{m}\left(t\right).$$
where $C_{m}$ is the time spent actually executing the microservice. $T_{m}$ is the time spent transmitting the service data to the server and the time spent sending the results back. $W_{m}$ is the time spent waiting in the queue. The computational time can be measured as follows:(2)$$C_{m}\left(t\right)=\mu_{m}\left(t\right)/{\mathrm{PR}}_{s}\left(t\right)$$
where μ represents the processing requirements of microservice, and ${\mathrm{PR}}_{s}$ represents the processing capabilities of the server. At any time *t*, network usage can be calculated using Equation (Equation 3):(3)$$\mathrm{NU}\left(t\right)=\delta *T_{m}\left(t\right)$$
where δ represents the tuple length.

Microservice can be offloaded to a surrogate device in L-R edge, R-R fog or conventional cloud. The transmission time taken to offload *i*th microservice to the server can be measured as:(4)$$T_{i}=X_{i}/B_{up}+R_{i}/B_{down}$$
where $X_{i}$ represents the size of the microservice, $R_{i}$ represents the size of the results, and $B_{up}$ and $B_{down}$ represents the bandwidths available for uploads and downloads, respectively.

Due to the complex dynamic scenario, the task arrival rate and execution rate should follow general distribution. The total time spent by each microservice *i* waiting for its execution on server *j* is computed as follows:(5)$$\sum_{i=1}^{n}\sum_{j=1}^{m}W_{ij}$$

We model our system using M/M/1. In this model, the first *M* represents the task arrival rate, which is governed by a Poisson distribution. The second *M* indicates that the task execution rate follows an exponential distribution, with a mean service time. The parameter *n* corresponds to the number of servers, each having unique capabilities and being available to process the incoming tasks. This waiting time is computed using queuing theory [26], considering the arrival rate and processing rate of the server to identify the congested server, as shown in Equation (Equation 6):(6)$$W_{m}\left(t\right)=\left(\sigma^{a}+\sigma^{s}\right)/\left(2\left(X^{a}-\mu^{s}\right)\right)$$
where $\mu^{s}$ and $\sigma^{s}$ are the standard mean and variance of the microservice execution time, respectively, and $X^{a}$ and $\sigma^{a}$ represent the mean and variance of its arrival intervals, respectively. The primary goal is to reduce the application execution response time for all incoming tasks. As a result, an optimal server is chosen that has enough computing resources to minimize computing time, considers the available bandwidth to minimize the transmission time, and measures the load on the server to minimize queueing delay. As a result, DMAP optimizes the task placement strategy for all microservices on available computing devices by assigning the greatest number of delay-sensitive tasks to the L-R ENs, reducing the latency and overall offloading time.

Let ${\mathrm{FT}}_{i}$ denote whether the task has been finished within the deadline constraint Equation (7b). If it is equal to one, the respective task has been completed before the deadline; otherwise, it has not been completed before the deadline. The above objectives and constraints are formulated as follows:
(7a)$$\begin{array}{c} \;Minimize:D_{m}\left(t\right) \end{array}$$
(7b)$$\begin{array}{c} Subject\;to:{\mathrm{FT}}_{t}=\left\{\begin{array}{c} 1,D_{m}\left(t\right)<Dmax_{m}\left(t\right)\\ 0,otherwise \end{array}\right. \end{array}$$
(7c)$$\begin{array}{c} \;Maximize:\sum_{i=1}^{m}{\mathrm{FT}}_{i}\left(t\right) \end{array}$$

In general, objectives for the scheduling problem are formulated in Equation (7). These include maximizing the number of finished tasks within deadline constraint Equation (7c) and minimizing the application execution delay in Equation ([Disp-formula FD7a-sensors-23-07142]).

#### 3.3.2. Analysis of Failure Rate

An application is considered as failed if its execution is not completed within the deadline. There are several possible reasons why this failure could have happened. In cases in which the device does not possess sufficient resources to finish the task within the deadline, experiences extended waiting period in the execution queue or faces network congestion, the finished time, as indicated in Equation (7b), becomes zero, signifying that the task could not be executed successfully before the deadline. The failure rate of an application at a specific time is given as follows:(8)FR(t)=η/N
where FR represents the failure rate at instance *t*, η represents the number of failed tasks and *N* represents the total number of tasks submitted for execution.

## 4. Proposed Microservices Application Scheduling

Based on the proposed n-tier fog-to-cloud architecture, we propose a microservice scheduling technique. The performance of the proposed algorithm is compared with baseline approaches, as discussed in Section 5. In the proposed DMAP, arriving microservices are inserted in a waiting queue, where they await their turn for execution. The computing capacities of each fog server in the L-R edge-tier are measured (line 7 of Algorithm 1) and are compared with the resource requirements of the microservice module represented by $CPU_{mn}$. They are then assigned to L-R EN that have enough computational resources for execution. Furthermore, different modules can be assigned to different ENs for execution. They can be executed in parallel. If the actual delay of the microservice is less than the threshold value, FT is updated to 1 and the service is marked to be executed successfully (line 9–11 of Algorithm 1). If the computing requirements of a module exceed the computing capabilities of EN in the L-R edge-tier, it is shifted to R-R fog-tier (line 14 of Algorithm 1). Algorithm 1 exhibits the workings of the microservice application placement in L-R edge in a three-tier fog-to-cloud environment.
**Algorithm 1** Workings of microservices application placement in L-R edge in three-tier fog-to-cloud**procedure** MICROSERVICE-PLACEMENT-LR() **for** each arriving application $A_{mn}$ **do**    insert all its microservices $m_{mn}$ in Queue *Q* for execution **end for** **for** each $m_{mn}$ in *Q* **do**    **for** each virtual machine $VM_{i}$ in L-R **do**      calculate $CPU_{vm}$      **if** $CPUv_{mi}\geq CPU_{mn}$ **then**         execute *m* on $VM_{i}$         **if** $FT_{m}=1$ **then**           mark the task as success         **end if**      **else**         MicroservicePlacmentRR($m_{mn}$)      **end if**    **end for** **end for****end procedure**

When the L-R edge-tier does not have enough resources to execute the microservice, the resources of the R-R fog tier in the three tier fog-to-cloud environment are considered. Algorithm 2 presents the steps involved in executing the microservice in the R-R fog tier in a three-tier fog-to-cloud environment. All the microservices are inserted in a queue waiting for their turn of execution. The computing capacity of each VM in the R-R fog tier is then measured. If the resource capacity ($CPU_{Vmi}$) of $VM_{i}$ is higher than the resource requirements of the microservice mn$CPU_{mn}$, the microservice is outsourced to $VM_{i}$; otherwise, it is outsourced to the cloud for its execution. If the service completes its execution within the specified delay requirement, FT is updated to one.
**Algorithm 2** Working of microservices application placement in R-R fog tier in three-tier fog-to-cloud technique 1:**procedure** MICROSERVICE-PLACEMENT-RR() 2: **for** each arriving microservice in HC $m_{mn}$ **do** 3:    insert all incoming microservices $m_{mn}$ in Queue *Q* to wait for their turn 4: **end for** 5: **for** each $m_{mn}$ in *Q* **do** 6:    **for** each virtual machine $VM_{i}$ in HC **do** 7:      calculate $CPU_{vm}$ 8:      **if** $CPUv_{mi}\geq CPU_{mn}$ **then** 9:         execute *m* on $VM_{i}$10:         **if** $FT_{m}=1$ **then**11:           mark task as success12:         **end if**13:      **else**14:         MicroservicePlacmentCloud($m_{mn}$)15:      **end if**16:    **end for**17: **end for**18:**end procedure**

## 5. Performance Evaluation

This section presents the empirical evaluation of DMAP in comparison to the baseline approaches proposed in the existing literature. We begin with explaining the configuration of the simulation, our execution of the methods, and the introduction of the DMAP technique we propose. Following that, we outline the examination of the outcomes. Finally, we summarize our findings.

### 5.1. Simulation Setup and Parameters

A simulation-based approach is considered to experiment with different parameter configurations. The proposed approach is evaluated in iFogSim2 [27], which is built on CloudSim simulator [28]. To evaluate the performance of the proposed technique, a mobile device can run a delay-sensitive application, EEG tractor beam game (EEGTBG) [18], or delay-tolerant application, video surveillance/object tracking (VSOT) [29], with a gradually increasing number of users. These applications are developed as a set of microservices that are executed separately on the containers by the simulators. The details of the application model, with their modules and interactions among them, can be found in [29,30]. We ensure the heterogeneity of microservices in terms of their respective computation costs as (300–900 MIPS) and bandwidth usage as (200-1500 bytes/packet). Additionally, a resource request of 100 and 200 per second is taken into consideration. The IoT simulation benchmarks reported in earlier simulation studies [18,29] are the foundation for all the aforementioned parametric values. Table 2 shows the overall configurations used in our simulations. B and I in Table 2 represent the busy and idle status of a device, respectively. The processing requirements of these two application modules are shown in Table 3.

### 5.2. Implementation of Algorithms

The efficiency of the proposed technique is investigated according to the following approaches:Local: In this approach, all services are executed locally on the mobile device.Cloud Only: In this approach, all services except the user interface module are executed on cloud data centers.Edgewards [32]: All services are executed on the EN if resources are available there; otherwise, they are offloaded upwards towards the central cloud.Volunteer-Supported (Vol-sup) [18]: This is an extension of the Edgewards approach, in which idle/underutilized resources of volunteer devices near FNs are used to execute services in parallel to address the issues of higher latency, network usage and energy consumption instead of offloading them to remote cloud servers. The nearby resources are considered horizontally before offloading services to remote cloud servers.

### 5.3. Delay Analysis

The loop delay for the proposed technique DMAP, Vol-sup, Edgewards, local-only, and cloud-only approaches is shown in Figure 4. All of them first execute their services at the nearby ENs, causing a comparable latency, until more users are added, and ENs can no longer support the additional users. The microservices are then moved to the cloud using the Edgewards approach, while they are moved to R-R by L-R according to the proposed technique, causing a slow increase in their respective latency.

Due to the core network’s capacity restriction, the latency of Edgewards approach increases with user 8 onwards for EEGTBG, as shown in Figure 4a. Likewise, the amount of data sent to the cloud grows as the number of users increases, eventually using all of the available bandwidth. Even though the cloud has an overwhelming amount of computing capacity, bandwidth restrictions create transmission delays that effect latency incurred using the Edgewards approach. The delay of the vol-sup approach starts increasing from user 24 onwards due to the bandwidth limitation of the core network as compared to DMAP, which can handle 40 users diligently. The DMAP tries to offload the services to the L-R edge-tier, and then H-R fog-tier, before offloading services to the central cloud. This results in offloading few services towards the cloud, as the maximum amount of resources available in fog tier and edge tier is utilized. There is no delay involved in executing application locally on the IoT device, but the energy consumption of local execution is comparatively high.

Figure 4b depicts a delay loop for delay-tolerant application (VSOT). Its value increases drastically as 14 users join the network for the Edgewards approach, whereas it can accommodate 24 users in the vol-sup approach, which is increased to 34 in the proposed technique. Hence, as depicted in Figure 4, it can be concluded that under normal to heavy load situations, which are reasonably predicted given the flood of data that are anticipated to be created under the IoT storm, our proposed technique DMAP performs better as compared to baseline approaches.

### 5.4. Network Usage Analysis

Figure 5 shows the network usage for EEGTBG and VSOT applications. The presence of the cloud with a propagation delay of 100 ms increases network usage. It is clearly depicted in Figure 5 that the bandwidth for the cloud-only approach is the worst, as microservices are moved to a distant cloud. With an increase in the user count, there is a corresponding rise in network load, leading to substantial network utilization. However, if we look closely at Figure 5a, we can see that the slope of network utilization for the Edgewards curve is steeper than that for the proposed approach due to the increased load. This spike also illustrates bandwidth constraint, which, as already discussed in the previous section, causes the delay in the Edgewards approach to increase drastically. On the other hand, because the delay of the IoT to edge tier link is just about 2 ms, the network utilization of the proposed technique is quite low. It increases drastically once 34 users join the network, which causes the resources of proximate servers to be exhausted and services to be offloaded to distant servers, compared to 16 for the Edgewards approach and 24 for the vol-sup approach, as shown in Figure 5b. As a result, when compared to the vol-sup and Edgewards approach, the network is busy for a shorter period of time. Therefore, as a result, the network is only momentarily active.

### 5.5. Energy Consumption of IoT Devices Analysis

Figure 6 compares the energy consumption rate of the proposed approach with local execution on IoT devices. The energy consumption of IoT devices is crucial; hence, it is taken into account when executing microservices. Only local execution is considered for comparison. This is because, when an offloading decision is made, IoT’s idle energy consumption rate is used. Therefore, other approaches are not taken into consideration for comparison. In contrast, when all services are executed locally, there is a higher rate of power usage.

The energy consumption of IoT devices, fog nodes and the cloud data center is shown in Figure 7. As is evident from Figure 7, the energy consumption of fog devices of the proposed technique is the highest compared to the rest of the approaches. This is because there is an attempt to deploy the maximum number of services at neighboring devices in the L-R edge tier or R-R fog tier. Figure 8 represents the total energy consumption of all devices in multi-tiered FC. By comparing the total energy consumption in this graph, it can be deduced that DMAP performs better than alternative strategies by decreasing the overall energy consumption of all devices within the multi-tiered FC architecture.

### 5.6. Service Failure Rate Analysis

Figure 9 illustrates the number of delay-sensitive tasks that are executed before their deadlines. It is evident that for a cloud-only approach, the delay is quite high and deadlines of tasks are not met. Task deadlines are mostly met; but, in some scenarios, they are not met due to limited resource constraints and communication overhead of local FNs, when the tasks need to be offloaded to a distant cloud. Hence, in most scenarios, DMAP fulfills the demands of tasks generated by IoT devices.

### 5.7. Summary

In this study, we focused on investigating the impact of outsourcing services to the R-R fog-tier. The offloading decision is made by the broker for its respective IoT devices. As the number of services increases, the delay and network utilization of the proposed technique DMAP proves to be superior as compared to the baseline approaches. This is verified by experimental results, as the proposed technique uses underutilized resources from the L-R edge tier and R-R fog tier, before offloading tasks to the cloud, lowering the energy consumption of all IoT devices. Since services are performed on nearby servers, network usage is also reduced. The proposed technique stands out from other methods due to these two significant reasons, as is evident from the analysis. It minimizes delay and network utilization by utilizing the resources of the R-R fog tier as well as the L-R edge tier before offloading services to the central cloud.

## 6. Conclusions and Future Work

This paper focuses on investigating the issue of microservices application scheduling in a multi-tier fog environment. Two fog tiers are implemented for this purpose: nearby L-R edge tier and R-R fog tier. Extensive simulations are conducted to compare the proposed technique with baseline approaches. Our findings reveal that the Edgewards method produces satisfactory outcomes for relatively small problem instances. However, when the infrastructure size and/or the number of services to be placed increases, the performance of the edgewards approach significantly declines. In contrast, all evaluated decentralized techniques exhibit superior scalability under these circumstances.

We intend to expand the proposed technique to include deep learning approaches in future work. The tasks can be forwarded to nearby FNs, and fog clustering techniques can be considered for further horizontal resource load balancing. Real-time implementation with fine-grained optimization can also be used to examine the effects of proposed decentralised microservices placement to validate these findings.

## Figures and Tables

**Figure 1 sensors-23-07142-f001:**
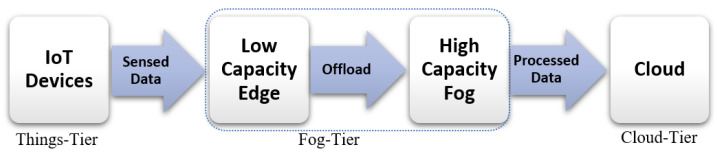
Three-tier fog-to-cloud architecture.

**Figure 2 sensors-23-07142-f002:**
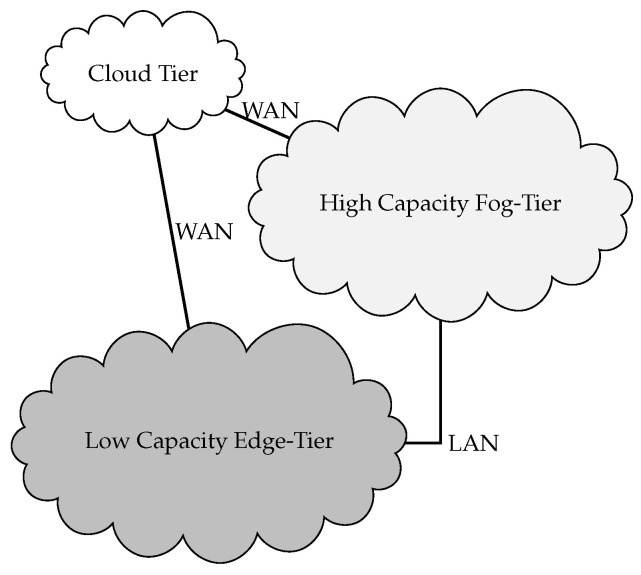
Three-tier fog-to-cloud architecture.

**Figure 3 sensors-23-07142-f003:**
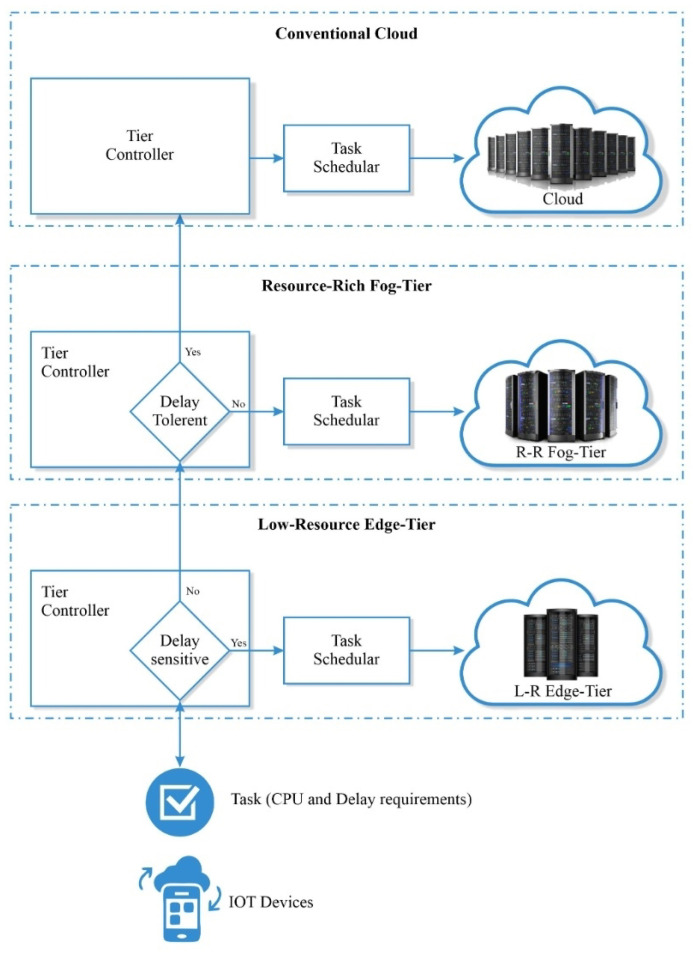
Multi-tiered fog computing architecture.

**Figure 4 sensors-23-07142-f004:**
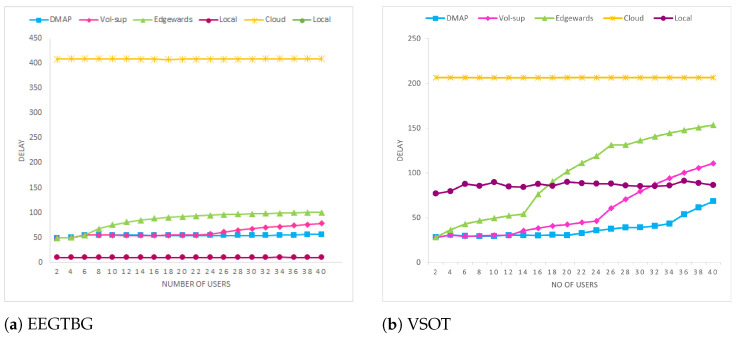
Delay Comparison.

**Figure 5 sensors-23-07142-f005:**
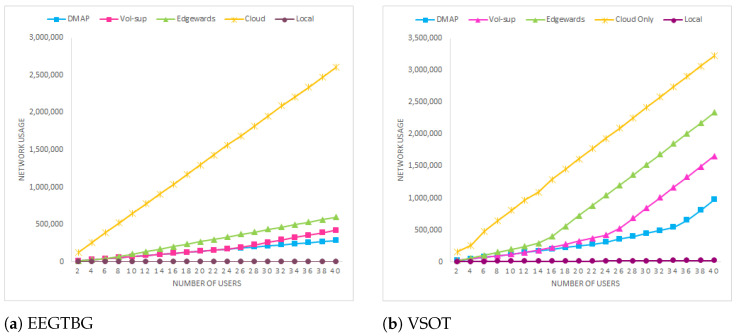
Network usage comparison.

**Figure 6 sensors-23-07142-f006:**
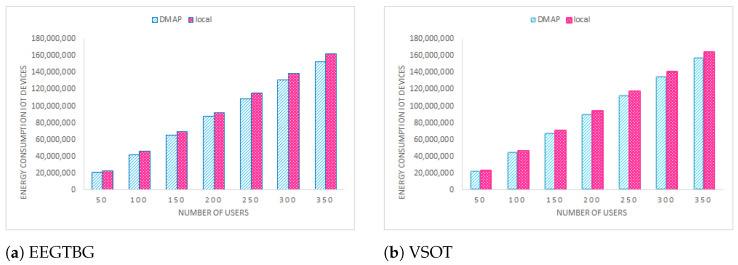
Energy consumption comparison.

**Figure 7 sensors-23-07142-f007:**
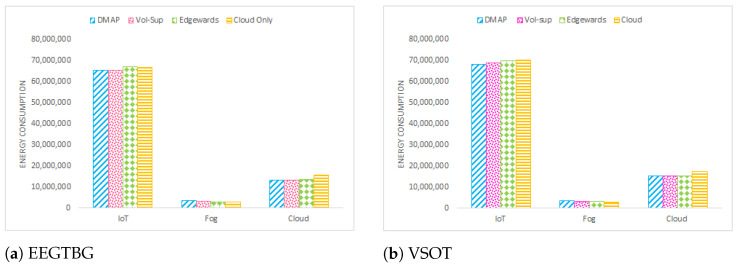
Energy consumption of tier landscape comparison.

**Figure 8 sensors-23-07142-f008:**
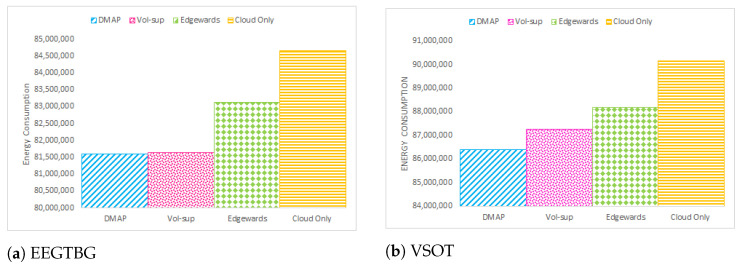
Total energy consumption comparison.

**Figure 9 sensors-23-07142-f009:**
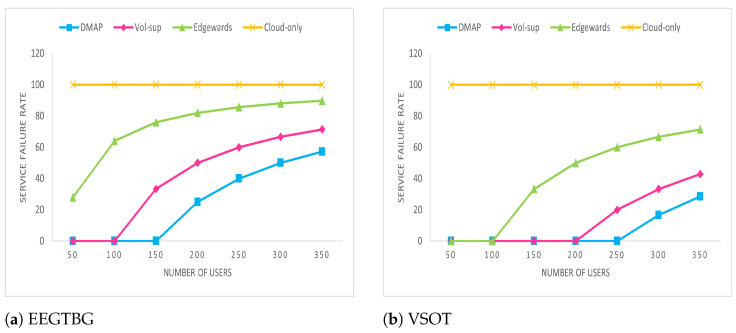
Service failure rate comparison.

**Table 1 sensors-23-07142-t001:** A qualitative comparison of the current application placement policies.

Technique	Simulator	IoT-Fog	Fog-Fog	Fog-Cloud	Failure Rate	Dynamic	Microservices	Scalability	Network Usage	Latency
[18]	iFogSim	✔	✘	✔	✘	✘	✘	✔	✔	✔
[19]	Java	✔	✘	✔	✘	✔	✘	✔	✘	✔
[20]	iFogsim	✔	✘	✔	✘	✘	✘	✔	✔	✔
[21]	Python	✘	✔	✔	✘	✔	✘	✘	✘	✔
[13]	CloudSim	✔	✔	✔	✔	✘	✘	✔	✘	✘
[22]	iFogSim	✔	✔	✔	✘	✔	✘	✔	✔	✔
[23]	iFogSim	✔	**-**	✔	✔	✔	✔	✔	✔	**-**
[24]	simulations	✔	✘	✘	✔	✔	✔	✔	✘	✔
DMAP	iFogSim2	✔	✔	✔	✔	✔	✔	✔	✔	✔

✔: Addressed, **-**: Partially Addressed, ✘: Not Addressed.

**Table 2 sensors-23-07142-t002:** Configuration parameters based on [18].

Device	CPU (GHz)	RAM (GB)	Processing (Mips)	Power (W)
Cloud	3.0	49,152	80,000,000	16 × 103 (B), 16 × 83.25 (I)
Fog	3.0	2.8	2048	107.339 (B), 83.433 (I)
Edge	3.0	40	44,800	107.339 (B), 83.433 (I)
IoT	1	40	44,800	87.53 (B), 82.44 (I)

**Table 3 sensors-23-07142-t003:** Processing requirements of EEGTBG and VSOT application modules based on [31].

EEGTBG				VSOT		
Object Detector	Motion Detector	Object Tracker	User Interface	Client	Concentration Calculator	Coordinator
550	300	300	200	200	350	100

## Data Availability

All supporting data will be provided on request.

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
