# Peer review of "Microservice Application Scheduling in Multi-Tiered Fog-Computing-Enabled IoT"

_sensors, 2023, doi:10.3390/s23167142_

Round 1

Reviewer 1 Report

In the manuscript, the authors have established an efficient multi-tiered fog-to-cloud architecture and proposed a decentralized resource mapping algorithm. Simulation experiments have been conducted based on the iFogSim2 simulator. The manuscript is well organized overall. However, some concerns need to be addressed.

1.     Some abbreviations are used without definition, such as FC and CDC.

2.     The quality of Figure 1 needs to be improved. It is kind of blurry.

3.     “We formulate the task offloading problem as an optimization problem that aims to minimize latency, network utilization and network utilization.” The sentence in Line 190 needs to be revised.

4.     The definition of Eq. (4) is informal, and its related explanation is missing. What does it refer to?

5.     Please double-check Eq. (6b) and (6c). Some details are incorrect.

6.     In terms of Algorithm 1, how split an application A_{mn} to microservices m_{mn}? What does “Calculate CPU_{Vm}” mean? What are the definitions of CPU_{Vmi} and CPU_{mn}?

7.     In Figure 4, does DMAP in Fig. 4 refer to the proposed technique? The abbreviation is used without any definition.

8.     Figure 5(a) is Delay. Should it be Network Usage?

9.     What is the definition of network usage?

Please carefully proofread the manuscript.

Author Response

Please find the attachment for the response to the reviewers comment

Reviewer 2 Report

1. It would be good to add latency and bandwidth as comparison items in table1.
2. Please review the optimal n of the n-tier edge.
3. Please mention the bandwidth of the performance metric.
4. Please evaluate the performance compared to the same iFogSim.

Author Response

(The authors gave the same response as above.)

Reviewer 3 Report

The paper primarily explores the use of microservice scheduling to address latency and resource utilization issues in fog-enabled Internet of Things (IoT) systems. However, there are still several problems that remain.

1.Detailed descriptions of the experimental procedures regarding the experiments in the paper are required.

2.The final presentation of the experimental results is unclear.

3.In the abstract, it is mentioned that "our proposed technique significantly improves the latency requirements of application execution, network bandwidth utilization, and energy consumption by 66.92%, 69.83%, and 4.16%." Is this improvement in comparison to the average of the 11 baselines or is it in comparison to a specific baseline?

The paper primarily explores the use of microservice scheduling to address latency and resource utilization issues in fog-enabled Internet of Things (IoT) systems. However, there are still several problems that remain.

1.Detailed descriptions of the experimental procedures regarding the experiments in the paper are required.

2.The final presentation of the experimental results is unclear.

3.In the abstract, it is mentioned that "our proposed technique significantly improves the latency requirements of application execution, network bandwidth utilization, and energy consumption by 66.92%, 69.83%, and 4.16%." Is this improvement in comparison to the average of the 11 baselines or is it in comparison to a specific baseline?

Author Response

(The authors gave the same response as above.)

Round 2

Reviewer 1 Report

Many thanks for the efforts of the authors. I have no further concerns.

The writing can be further improved.

Reviewer 3 Report

Good

Good